# SEQUENTIAL DATA GENERATION WITH GROUPWISE DIFFUSION PROCESS

## ABSTRACT

We present the Groupwise Diffusion Model (GDM), which divides data into multiple groups and diffuses one group at one time interval in the forward diffusion process. GDM generates data sequentially from one group at one time interval, leading to several interesting properties. First, as an extension of diffusion models, GDM generalizes certain forms of autoregressive models and cascaded diffusion models. As a unified framework, GDM allows us to investigate design choices that have been overlooked in previous works, such as data-grouping strategy and order of generation. Furthermore, since one group of the initial noise affects only a certain group of the generated data, latent space now possesses group-wise interpretable meaning. We can further extend GDM to the frequency domain where the forward process sequentially diffuses each group of frequency components. Dividing the frequency bands of the data as groups allows the latent variables to become a hierarchical representation where individual groups encode data at different levels of abstraction. We demonstrate several applications of such representation including disentanglement of semantic attributes, image editing, and generating variations.

## 1 INTRODUCTION

Diffusion models (Ho et al., 2020; Song & Ermon, 2019; Song et al., 2020; Sohl-Dickstein et al., 2015; Dhariwal & Nichol, 2021) are recently popularized generative models that have shown impressive capabilities in modeling complex datasets. Diffusion models generate data by inverting the noise-adding forward process, which is one of the most critical design choices as the reverse generative process is merely its time-reversed counterpart. As such, extensive research has explored alternatives to the standard noise-adding forward process, demonstrating that other forms of degradation—such as blurring, masking, downsampling, snowification, or pre-trained neural encoding—can also be used for the forward process of diffusion models (Rissanen et al., 2022; Lee et al., 2022; Hoogeboom & Salimans, 2022; Daras et al., 2022; Gu et al., 2022; Bansal et al., 2022).

In this paper, we present the Groupwise Diffusion Model (GDM), which uses a new type of forward process that divides data into multiple groups and diffuses one group at once. As a time reversal of the group-wise noising process, the generative process of GDM synthesizes data sequentially from one group at one time interval, leading to several interesting properties.

**Unified framework**   Currently, diffusion models are state-of-the-art in generating continuous signals including images (Dhariwal & Nichol, 2021; Rombach et al., 2022; Saharia et al., 2022) and videos (Ho et al., 2022b), while the autoregressive (AR) models excel at generating sequence of discrete data such as languages (OpenAI, 2023). One advantage of AR models is that they can leverage prior knowledge when data have a natural ordering. However, the computational costs of AR models scale linearly to the length of the sequence, while diffusion models can decouple the dimensionality of data and the number of sampling steps. The unified framework for the two models has yet to be explored.

We show that GDM serves as a bridge between diffusion models and certain forms of AR and cascaded diffusion models (Ho et al., 2022a). As a unified framework, GDM allows us to investigate design choices for generative models that have been overlooked in previous work, such as data-grouping strategy and order of generation.

**Interpretable latent space**   Diffusion models are trained by learning the drift of an ODE that yields the same marginal distributions as a given forward diffusion process. In such a viewpoint, diffusion models learn a one-to-one mapping between data $\boldsymbol{x} \in \mathbb{R}^d$ and noise $\boldsymbol{z} \in \mathbb{R}^d$ defined via ODE, assigning a unique latent variable $\boldsymbol{z}$ to each data point as in other invertible models.

One notable characteristic of GDM is that changing one group of $\boldsymbol{z}$ affects *only certain elements of $\boldsymbol{x}$*, and the relationship between the groups of latent variables and the data elements they affect is set in advance. As a result, latent space now possesses group-wise interpretable meaning (i.e. we know which elements of initial noise affect which parts of data).

**Hierarchical representation**   We can further extend GDM to the frequency domain –dubbed GDM-F– where the forward process sequentially diffuses each group of frequency components. Each group of latent variables (or latent group) thus selectively influences specific frequency bands of data, providing a hierarchical representation where individual groups encode data at different levels of abstraction. We demonstrate several applications of such representation on image datasets, including disentanglement of semantic attributes, image editing, and generating variations.

## 2   BACKGROUND ON DIFFUSION MODELS

Diffusion models are generative models that synthesize data by simulating a reverse-time SDE of a given diffusion process, which is often converted to the marginal-preserving ODE for efficient sampling. From a rectified flow (Liu et al., 2022; Liu, 2022) (or similarly, stochastic interpolant (Albergo et al., 2023; Albergo & Vanden-Eijnden, 2022)) perspective, the forward diffusion process for variance-preserving diffusion models (Song et al., 2020) can be viewed as a nonlinear interpolation between data $\boldsymbol{x} \sim p(\boldsymbol{x})$ and noise $\boldsymbol{z} \sim p(\boldsymbol{z})$:

$$\boldsymbol{x}_t(\boldsymbol{x}, \boldsymbol{z}) = \alpha(t)\boldsymbol{x} + \sqrt{1 - \alpha(t)^2}\boldsymbol{z}, \tag{1}$$

where $\alpha(t)$ is a nonlinear function of $t$ with $\alpha(0) = 1$ and $\alpha(1) \approx 0$, and $p(\boldsymbol{z}) = \mathcal{N}(\boldsymbol{0}, \mathbf{I})$. Some recent works (Lipman et al., 2022; Liu et al., 2022) instead use the linear interpolation

$$\boldsymbol{x}_t(\boldsymbol{x}, \boldsymbol{z}) = (1 - t)\boldsymbol{x} + t\boldsymbol{z}, \tag{2}$$

for a constant velocity (given $\boldsymbol{x}$ and $\boldsymbol{z}$) that has better sampling-time efficiency.

Diffusion models are trained by minimizing a time-conditioned denoising autoencoder loss (Vincent, 2011)

$$\min_{\boldsymbol{\theta}} \mathbb{E}_{t \sim U(0,1)} \mathbb{E}_{\boldsymbol{x},\boldsymbol{z}}[\lambda(t)||\boldsymbol{x} - \boldsymbol{x}_{\boldsymbol{\theta}}(\boldsymbol{x}_t(\boldsymbol{x}, \boldsymbol{z}), t)||_2^2], \tag{3}$$

where $\lambda(t)$ is a weighting function, and $\boldsymbol{x}_{\boldsymbol{\theta}}(\cdot)$ is a neural network parameterized by $\boldsymbol{\theta}$. Instead of predicting $\boldsymbol{x}$, Liu et al. (2022) directly train a vector field $\boldsymbol{v}_{\theta}(\boldsymbol{x}_t, t)$ to match the time derivative $\frac{\partial \boldsymbol{x}_t}{\partial t}$ of Eq. (2) by optimizing

$$\min_{\boldsymbol{\theta}} \mathbb{E}_{t \sim U(0,1)} \mathbb{E}_{\boldsymbol{x},\boldsymbol{z}}[||(\boldsymbol{z} - \boldsymbol{x}) - \boldsymbol{v}_{\boldsymbol{\theta}}(\boldsymbol{x}_t(\boldsymbol{x}, \boldsymbol{z}), t)||_2^2]. \tag{4}$$

Inference and sampling are done by solving the following ODE (Liu et al., 2022) forward and backward in time, respectively:

$$d\boldsymbol{z}_t = \boldsymbol{v}_{\boldsymbol{\theta}}(\boldsymbol{z}_t, t)dt, \tag{5}$$

where $dt$ is an infinitesimal timestep.

## 3   GROUPED LATENT FOR INTERPRETABLE REPRESENTATION

In Section 3.1, we introduce the concept of *groupwise diffusion*, where data is segmented into multiple groups, and each group is diffused within the dedicated time interval. Following this, Section 3.2 demonstrates how our Groupwise Diffusion Model (GDM) serves as a generalization of certain instances of autoregressive models and cascaded diffusion models Ho et al. (2022a). We then show the group-wise interpretability of GDM's latent space in Sec. 3.3 and its implication in the frequency domain in Sec. 3.4 . Related work is deferred to Appendix D.

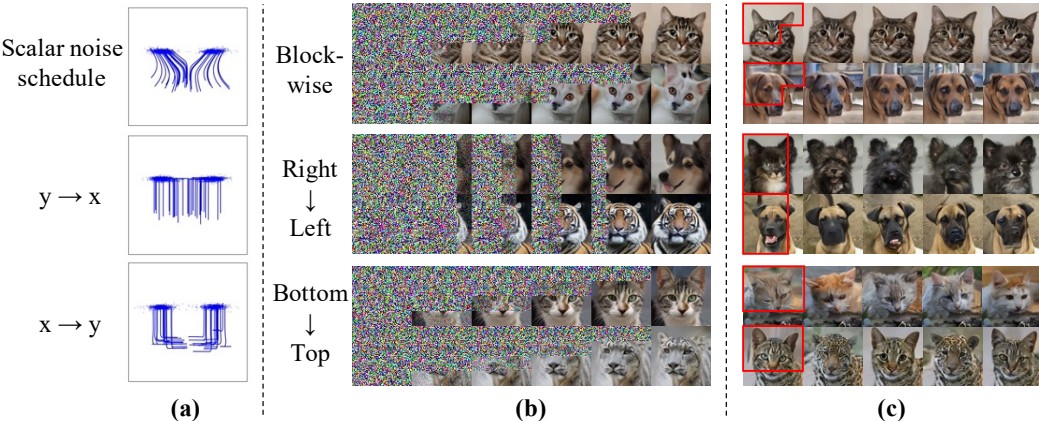

**(a)** **(b)** **(c)**

Figure 1: Generative process of GDM on (a) 2D Gaussian mixture and (b) AFHQ $64 \times 64$ datasets. While the previous method with scalar noise schedule denoises all elements simultaneously, our model generates (a) each element or (b) a group of pixels sequentially. (c) Varying each latent group only affects the corresponding group of pixels (indicated by red boxes) while others remain unchanged.

### 3.1 PER-GROUP NOISE SCHEDULING

Unlike previous diffusion models where all elements of data have the same noise schedule, we divide data $\boldsymbol{x} \in \mathbb{R}^d$ into $k \in \{1, ..., d\}$ disjoint groups and assign different noise schedules for each group. Specifically, we divide the indices $\{1, ..., d\}$ into a partition $\{S_j\}_{j=1}^k$ where $\sum_j |S_j| = d$. $S_j$ is used for determining membership of $\boldsymbol{x}_i$ to $j$-th group by testing $i \in S_j$. Then, we let the forward process diffuse elements of each partition within the assigned time interval.

To define separate noise schedules for each group, it is necessary to extend the previous scalar-valued noise schedule to a more general form. Lee et al. (2022) introduces several generalizations of previous diffusion models, including the utilization of a matrix-valued function $\mathbf{A}(t)$ instead of a scalar-valued function $\alpha(t)$ for the coefficient of the interpolation between data and noise. Applying this to the rectified flow, we extend Eq. (2) to

$$\boldsymbol{x}_t(\boldsymbol{x}, \boldsymbol{z}) = \mathbf{A}(t)\boldsymbol{x} + (\mathbf{I} - \mathbf{A}(t))\boldsymbol{z}, \tag{6}$$

where $\mathbf{A}(t)$ is a diagonal matrix satisfying $\mathbf{A}(0) = \mathbf{I}$ and $\mathbf{A}(1) \approx \mathbf{0}$. We build upon Eq. (2) for convenience, and other interpolations such as Eq. (1) are equally applicable. Now, the training objective becomes

$$\min_{\boldsymbol{\theta}} \mathbb{E}[|| \underbrace{\mathbf{A}'(t)(\boldsymbol{x} - \boldsymbol{z})}_{=\frac{\partial \boldsymbol{x}_t}{\partial t}} - \boldsymbol{v}_{\boldsymbol{\theta}}(\boldsymbol{x}_t(\boldsymbol{x}, \boldsymbol{z}), \mathbf{A}(t))||_2^2], \tag{7}$$

where $\mathbf{A}'(t) = \frac{\partial \mathbf{A}(t)}{\partial t}$. Note that now $\boldsymbol{v}_{\boldsymbol{\theta}}(\cdot)$ receives $\mathbf{A}(t)$ instead of $t$, which is concatenated to noised data $\boldsymbol{z}_t$ in our implementation. This allows us to train a single model compatible with multiple noise schedules. However, directly predicting $\mathbf{A}'(t)(\boldsymbol{x} - \boldsymbol{z})$ might be challenging for a neural network since it has to infer $\mathbf{A}'(t)$ from $\mathbf{A}(t)$. Instead, we find it beneficial to define $\mathbf{u}_{\boldsymbol{\theta}}(\cdot) \triangleq \mathbf{A}'(t)^{-1}\boldsymbol{v}_{\boldsymbol{\theta}}(\cdot)$ and optimize the unweighted variant of Eq. (7) as in Lee et al. (2022):

$$\min_{\boldsymbol{\theta}} \mathbb{E}[||(\boldsymbol{x} - \boldsymbol{z}) - \mathbf{u}_{\boldsymbol{\theta}}(\boldsymbol{x}_t(\boldsymbol{x}, \boldsymbol{z}), \mathbf{A}(t))||_2^2]. \tag{8}$$

We draw samples by solving

$$d\boldsymbol{z}_t = \underbrace{\mathbf{A}'(t)\mathbf{u}_{\boldsymbol{\theta}}(\boldsymbol{z}_t, \mathbf{A}(t))}_{=\boldsymbol{v}_{\boldsymbol{\theta}}(\boldsymbol{z}_t, \mathbf{A}(t))} dt. \tag{9}$$

This formulation allows us to define a different noise schedule for each element of data, therefore providing more flexibility in designing diffusion models. For $i \in S_j$, the $i$-th diagonal entry of $\mathbf{A}(t)$

is defined as follows:

$$\mathbf{A}(t)_{ii} = \begin{cases} 1, & (0 \leq t \leq t_{\text{start}_j}) \\ \frac{1}{t_{\text{start}_j} - t_{\text{end}_j}} t - \frac{1}{t_{\text{start}_j} - t_{\text{end}_j}} t_{\text{end}_j}, & (t_{\text{start}_j} < t \leq t_{\text{end}_j}) \\ 0, & (t > t_{\text{end}_j}) \end{cases} \tag{10}$$

Here, the elements of $j$-th group are diffused into noise during the interval $[t_{\text{start}_j}, t_{\text{end}_j}]$. Note that when $k = 1$, $\mathbf{A}(t) = (1 - t)\mathbf{I}$, and Eq. 6 becomes Eq. (2). Each $t_{\text{start}_j}$ and $t_{\text{end}_j}$ are predefined such that the time intervals of each group do not overlap with each other. That is, only one group of elements is diffused (and therefore generated) within a certain time interval. For example, we can make diffusion models generate data from one element at once (Fig. 1 (a)) or from one grid at once (Fig. 1 (b)) using this per-group noise scheduling.

## 3.2 GDM GENERALIZES AUTOREGRESSIVE MODELS AND CASCADED DIFFUSION MODELS

We find that two popular generative models, autoregressive (AR) generative models and cascaded diffusion models (CDM) (Ho et al., 2022a), are special cases of our GDM with certain grouping strategies. We only provide the main results here, and a detailed explanation is deferred to Appendix C.

**Proposition 3.1.** *Define autoregressive models as $p_{\boldsymbol{\theta}}(\boldsymbol{x}) = \prod_{i=1}^{d} p_{\boldsymbol{\theta}}(x_i|x_{<i})$ where $p_{\boldsymbol{\theta}}(x_i|x_{<i}) = \mathcal{N}(f_{\boldsymbol{\theta}}(x_{<i}), \sigma^2 I)$ with a neural network $f_{\boldsymbol{\theta}}(\cdot)$ and a constant $\sigma$. This is a special case of GDM where the number of groups $k$ is equal to the data dimension $d$, and the number of steps $N_j$ for each group $j$ is set to $1$.*

One superiority GDM possesses over AR models is the flexibility of choosing the number of groups $k$ and the number of steps of each group $N_j$. For example, some groups of pixels can be generated in parallel without harming quality using sampling steps of more than one but substantially fewer than the total number of pixels in that group. For example, in Block-wise grouping in Fig. 1, each group has $21 \times 21$ pixels but less than 20 steps are used for each group. An extreme case is the vanilla diffusion model, where all pixels are generated simultaneously using a large number of sampling steps.

**Proposition 3.2.** *GDM generalizes cascaded diffusion models without low-pass filtering.*

CDMs learn the generative process in low-to-high resolution strategy where low-resolution data are subsampling of high-resolution data. Here, we assume that no low-pass filtering is used before subsampling. Since each group of pixels is generated sequentially in CDMs, this exact behavior can be also modeled by our method.

The advantage of GDM over AR models and CDMs is that our method is not restricted to their grouping strategy and order of generation. Therefore, we can explore alternative choices of grouping strategies rather than per-element generation (AR) and progressive scale generation (CDM) to maximize sample quality. Due to the computational complexity, we explore only CDMs.

## 3.3 GROUP-WISE INTERPRETABLE LATENT SPACE

Since each latent group contributes to only a certain phase of the generative process, the role of each group is explicitly predetermined by setting $S_j$, $t_{\text{start}_j}$, and $t_{\text{end}_j}$.

**Proposition 3.3.** *Let $\boldsymbol{x}_{\boldsymbol{\theta}^*}(\boldsymbol{x}_t, t)$ be the optimal denoiser. For $j$ with $t_{end_j} < t$, $\frac{\partial}{\partial(\boldsymbol{x}_t)_i} \boldsymbol{x}_{\boldsymbol{\theta}^*}(\boldsymbol{x}_t, t) = \mathbf{0}$ for all $i \in S_j$.*

*Proof.* Let $\mathbf{P}_1 = \{(\boldsymbol{x}_t)_i\}_{i \in S_j}$ and $\mathbf{P}_2 = \{(\boldsymbol{x}_t)_i\}_{i \notin S_j}$. Based on the definition of $\mathbf{A}(t)$ in Eq. (10), $\mathbf{P}_1$ is independently sampled noise as $t_{\text{end}_j} < t$, so $\boldsymbol{x}_{\boldsymbol{\theta}^*}(\boldsymbol{x}_t, t) = \mathbb{E}[\boldsymbol{x}|\boldsymbol{x}_t] = \mathbb{E}[\boldsymbol{x}|\mathbf{P}_1, \mathbf{P}_2] = \mathbb{E}[\boldsymbol{x}|\mathbf{P}_2]$. Therefore, $\frac{\partial}{\partial(\boldsymbol{x}_t)_i} \boldsymbol{x}_{\boldsymbol{\theta}^*}(\boldsymbol{x}_t, t) = \mathbf{0}$ if $(\boldsymbol{x}_t)_i \in \mathbf{P}_1$. $\qquad \square$

Proposition (3.3) implies that during sampling, the optimal denoiser ignores the elements that have not started to be generated yet. The fact that a latent group only contributes certain elements of data provides an interpretable meaning to each group. For instance, varying each latent group only affects a specific region of the generated images, as shown in Fig. 1 (c).

### 3.4 EXTENSION TO FREQUENCY DOMAIN

Our generative process can be extended to the frequency domain, where the components of each frequency band are sequentially generated. This is enabled by another generalization proposed in Lee et al. (2022), the choice of coordinate systems where diffusion is performed. Using this, Eq. (6) is further generalized to

$$\bar{\boldsymbol{x}}_t(\bar{\boldsymbol{x}}, \bar{\boldsymbol{z}}) = \mathbf{A}(t)\bar{\boldsymbol{x}} + (\mathbf{I} - \mathbf{A}(t))\bar{\boldsymbol{z}}, \tag{11}$$

where $\bar{\boldsymbol{x}} = \mathbf{U}^T\boldsymbol{x}$ and $\bar{\boldsymbol{z}} = \mathbf{U}^T\boldsymbol{z}$ for an orthogonal matrix $\mathbf{U}$. Depending on the choice of basis $\mathbf{U}$, GDM can be extended to the frequency domain, which we hereafter denote as GDM-F (F means frequency).

While the domain is changed, the training loss and generative ODE remain the same as GDM. However, we find that sample quality degrades severely if the input and output of a model are from the frequency domain. This is expected since convolutional networks are not intended for such inputs and outputs. We instead parameterize our network such that its inputs and outputs are in pixel space and the frequency transformation is performed outside the network. See Appendix A.1 for more details. We set $t_{\text{start}_j}$ and $t_{\text{end}_j}$ such that images are generated sequentially from low frequencies to high frequencies.

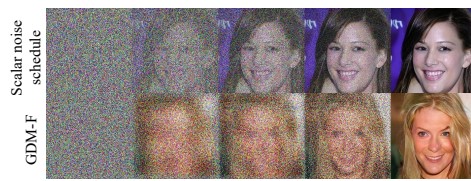

Figure 2: Visualization of generative processes of previous diffusion model with scalar noise schedule and GDM-F.

See Fig. 2 for visualization of generative processes. In contrast to a vanilla diffusion model where some details like eyes and mouth are visible in the earlier phase, GDM-F generates an image in a strictly hierarchical manner from low to high frequency.

By extending our method to the frequency domain, we can now divide an image into $k$ frequency bands and assign each latent group to each band. Therefore, we obtain representation organized into $k$-level hierarchy from low frequency to high frequency, where each group corresponds to a certain level of abstraction.

## 4 EXPERIMENTS

### 4.1 DESIGN CHOICES FOR MAXIMIZING SAMPLE QUALITY

**Grouping strategy** As a generalization of AR models and CDMs, our GDM opens up other choices for grouping strategies. Tab. 1 compares generative performances of the following grouping strategies: *Bottom→Top*, *Right→Left*, *CDM* and *CDM-inv*. *CDM* denotes the grouping strategy of the cascaded diffusion models. *CDM-inv* follows the same grouping strategy but in the inverse order. The

| Bottom→Top | Right→Left | CDM | CDM-inv |
|---|---|---|---|
| **12.63** | 13.14 | 14.03 | 13.70 |

Table 1: FID10K of each grouping strategy measured in cifar-10 dataset. We use Euler's solver with 128 sampling steps and $k = 2$ for all strategies.

best Frechet Inception Distance (FID, Heusel et al. (2017)) is achieved when using *Bottom→Top* grouping strategy.

**Order of generation** We further investigate the effect of the order of generation under the same *Block-wise* grouping strategy. As shown in Fig. 3, FID largely varies depending on the order in which each block is generated.

We note that the combinations of grouping strategy and order of generation considered here merely represent a fraction of numerous possible options, so it could not be claimed that a particular setting is optimal. Nonetheless, these results demonstrate that the way data is partitioned and the sequence in which it is generated – a technique commonly employed when generating high-dimensional data – largely affect generative performance.

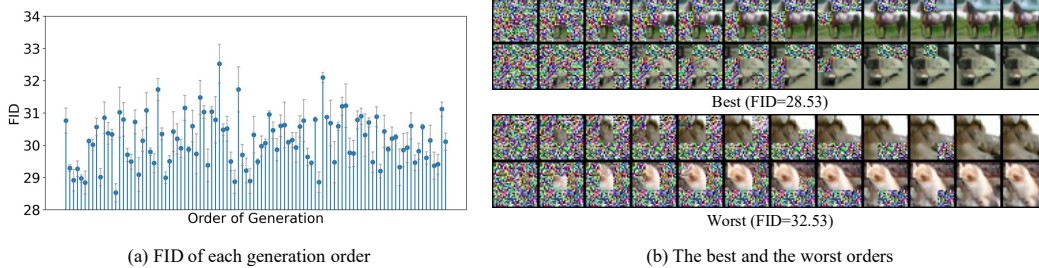

| (a) FID of each generation order | (b) The best and the worst orders |

Figure 3: (a) FID 5K (averaged over three runs) of 100 random orders of generation measured in cifar-10 dataset. Error bars denote standard deviation. Block-wise grouping with $k = 9$ is used. (b) Generative processes of the best and the worst orderings are displayed. We use the identical model for all orders of generation.

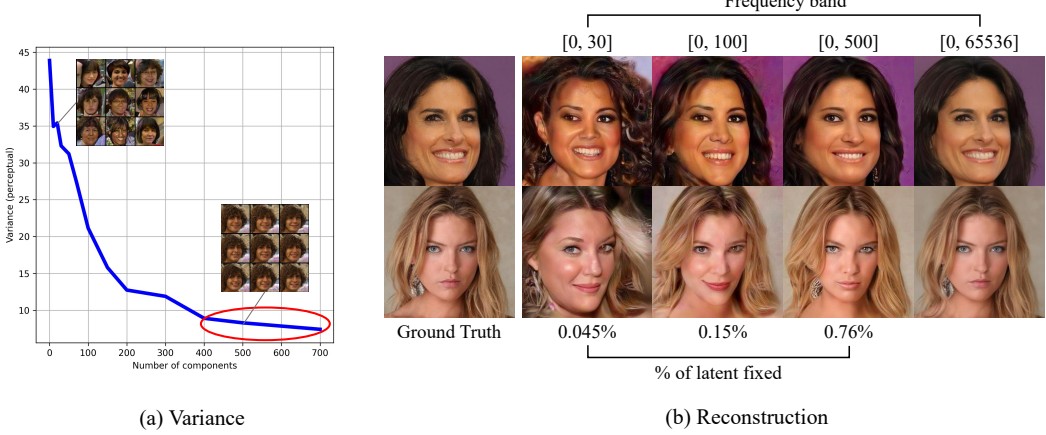

| (a) Variance | (b) Reconstruction |

Figure 4: (a) Variance of images generated by fixing latent variables in a specific band, and (b) reconstruction results using a subset of latent variables. The indices are sorted in ascending order starting from the lowest frequency band.

## 4.2 FINDING SEMANTIC BAND FOR GDM-F

GDM-F allows us to obtain a representation where each latent group corresponds to specific frequency bands. In order to endow with a useful hierarchy, we need to figure out which frequency band contains abstract (or semantic) information of data. In Fig. 4 (a), we fix the low-frequency components of the latent variables (the rest components are randomly sampled) and measure the perceptual variance of generated images $\mathbb{E}[||\phi(x) - \mathbb{E}[\phi(x)]||^2]$, where $\phi(\cdot)$ is VGG-19 feature extractor. As we increase the number of the fixed components, generated samples become semantically similar to each other. We find that fixing roughly 400 to 700 components (indicated by a red ellipse) among 4096 is sufficient to synthesize semantically consistent images.

In Fig. 4 (b), we encode real images and reconstruct them using a subset of the encoded latent variables. We can see that the latent variables in [0, 500] band, which accounts for only 0.76% of the total number of elements, suffice to faithfully reconstruct the ground truth images.

## 4.3 THE ROLE OF EACH LATENT GROUP OF GDM-F

Based on the above observations, it is reasonable to divide latent variables into at least two groups to obtain a useful hierarchy, one for semantics and another for details. Here, we further divide latent variables into more than two groups and see if it allows for more fine-grained controllability. Specifically, we divide a latent vector into 4 groups (i.e., $k = 4$) in the frequency domain. For the dataset of $256 \times 256$ resolution, we set $S_1 = \{0, ..., 29\}, S_2 = \{30, ..., 99\}, S_3 = \{100, ..., 499\}$, and $S_4 = \{500, ..., 256 \times 256 - 1\}$.

Fig. 5 shows the images synthesized by interpolating the elements of latent vectors corresponding to each frequency band on AFHQ $256 \times 256$ and CelebA-HQ $256 \times 256$ datasets. When interpo-

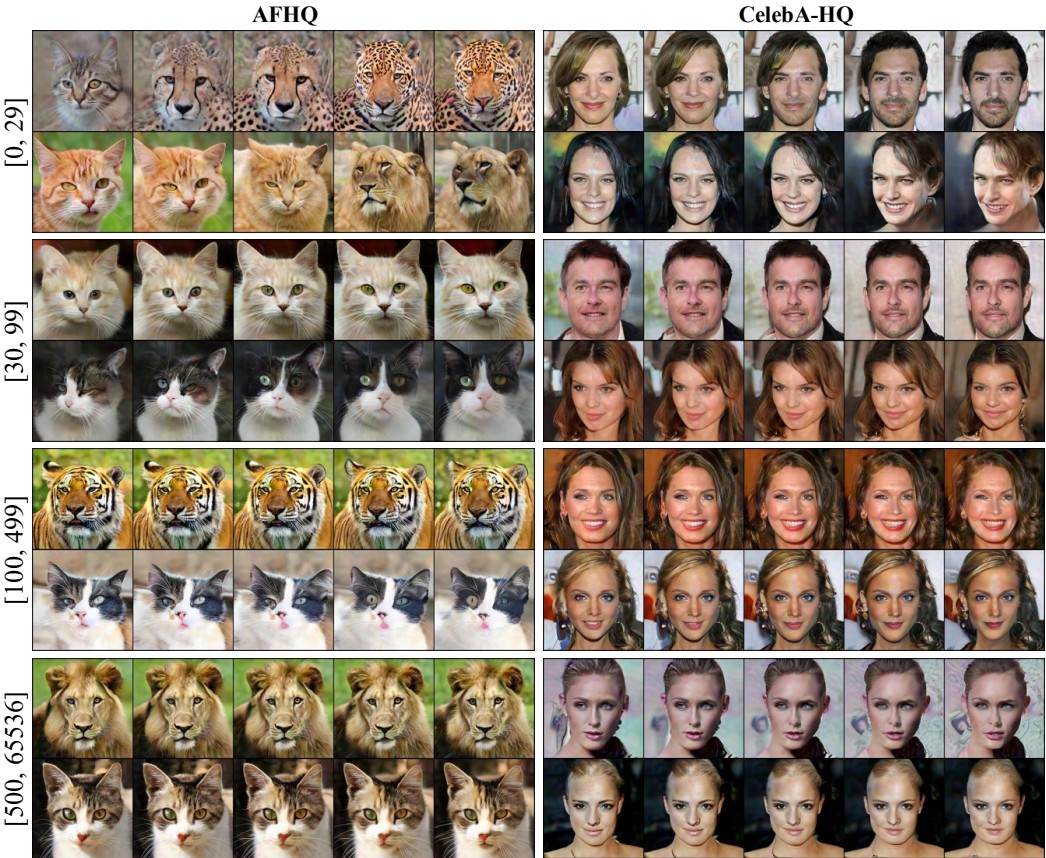

Figure 5: Generated images by interpolating the latent group for each frequency band while others fixed.

lating the coarsest frequency band [0, 29], high-level attributes such as gender, azimuth, or animal class are transformed smoothly. Conversely, interpolating the elements of [100, 499] band results in variation in fine attributes like facial expression. We can see that the elements in [500, 65536] do not change the content of images in a meaningful way, indicating that these elements control only minor attributes in images.

## 4.4 FACTORS OF VARIATION

Currently, most of the state-of-the-art disentanglement methods are based on VAEs(Kim & Mnih, 2018; Higgins et al., 2016; Chen et al., 2018). Recent studies focus on minimizing Total Correlation (TC) while maximizing reconstruction quality to force the marginal distribution of features independent. However, as TC is generally intractable, it needs to be approximated via minibatch statistics (Chen et al., 2018) or mini-max optimization (Kim & Mnih, 2018; Yeats et al., 2022).

It is noteworthy that since $p(\bar{z}) = \mathcal{N}(\mathbf{0}, \mathbf{I})$, the inference distribution (the marginal of features obtained by solving Eq. 5) of GDM-F has zero TC in the optima.

**Remark 4.1.** *In optima of Eq.* (4), *Eq.* (5) *maps between $p(\boldsymbol{x})$ and $p(\boldsymbol{z})$. See Theorem 3.3 in Liu et al. (2022) for the proof. As a consequence, the inference distribution has zero Total Correlation in the optima if we define $p(\boldsymbol{z})$ such that $p(\boldsymbol{z}) = \prod_i p(\boldsymbol{z}_i)$. This also holds for more general interpolations including the ones used in GDM.*

Moreover, unlike standard diffusion models, only a few elements in the semantic band of $\bar{z}$ capture the most information of data as we have seen in Sec. 4.2. We find that such a statistically independent semantic feature effectively captures high-level attributes like azimuth, hair length, gender, etc. at each element. Fig. 6 shows that manipulating *a single element* in the latent vector of the lowest frequency band results in a change in a single high-level attribute of images.

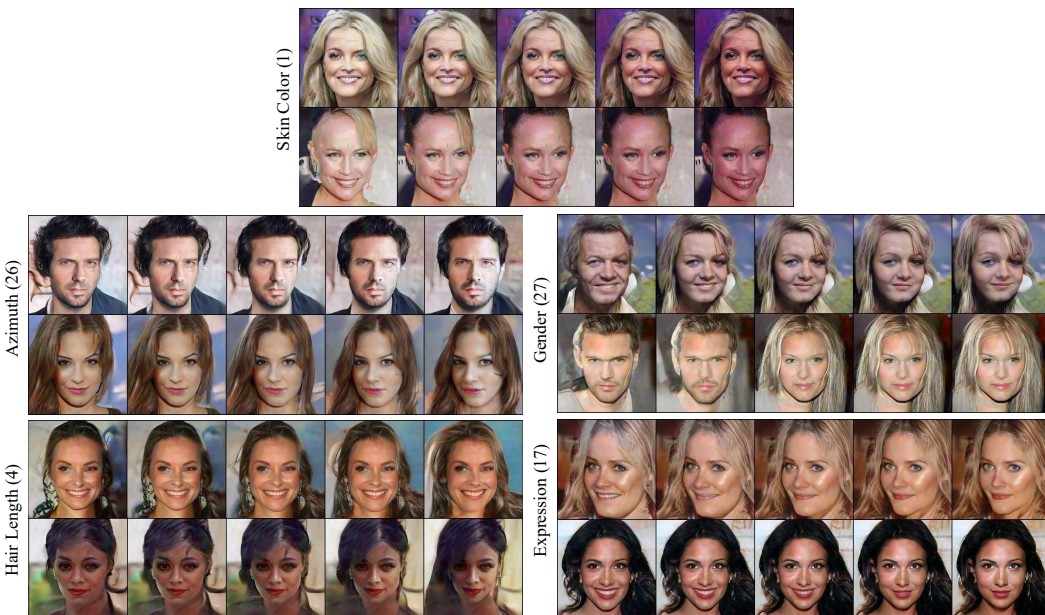

Figure 6: Synthesis results of GDM-F, traversing a single latent variable corresponding to the lowest frequency band over $[-3, 3]$ range. The numbers in parentheses denote the indices of the element traversed.

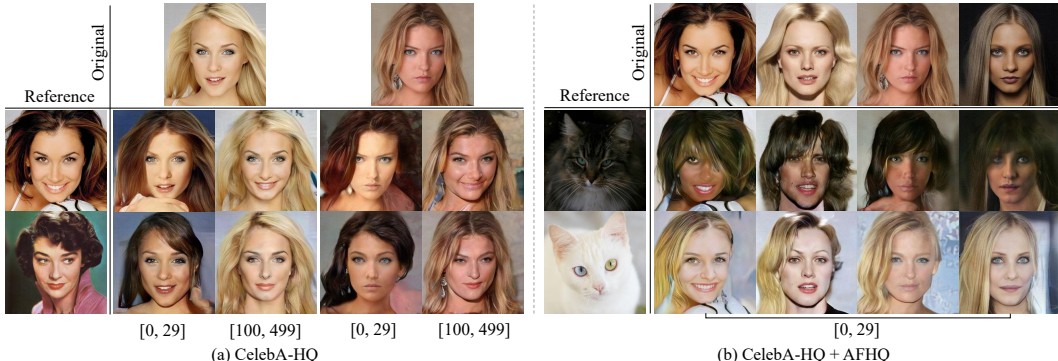

Figure 7: Synthesis results of GDM-F by mixing the latent code of two images. We replace a group of the latent code from *Original* with that of *Reference* in each column. The images can be either from (a) the same dataset or (b) different datasets. The parentheses indicate the frequency bands replaced.

## 4.5 IMAGE EDITING

Since the inference of GDM-F is invertible, our method can be applied to editing real images. This leads us to applications such as image mixing, attribute manipulation, and image variation.

**Image mixing**  Fig.7 demonstrates mixing the style of two images by swapping their latent code of specific frequency bands. In (a), swapping the coarse latent group $[0, 29]$ transfers high-level attributes, while swapping the fine latent group $[100, 499]$ alters facial expressions. (b) illustrates mixing images from different datasets similarly to DDIB(Su et al., 2022). We train our model on both datasets, encode images from each, and swap latent codes in the $[0, 29]$ range. Unlike DDIB, our organized latent variables allow us to selectively transfer the attributes of a certain level of abstraction, whereas DDIB's unstructured latent variables lack this capability.

**Attribute manipulation**  Fig. 8 illustrates that we can control a single attribute of input images by manipulating one element of the obtained representation. Specifically, we first obtain a feature of an

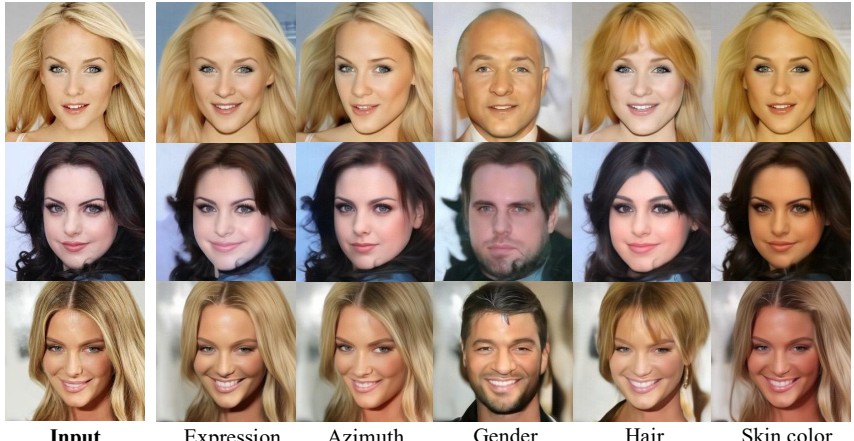

**Input**      Expression     Azimuth     Gender     Hair     Skin color

Figure 8: Real image editing results of GDM-F. We manipulate only a single element of the coarsest latent group.

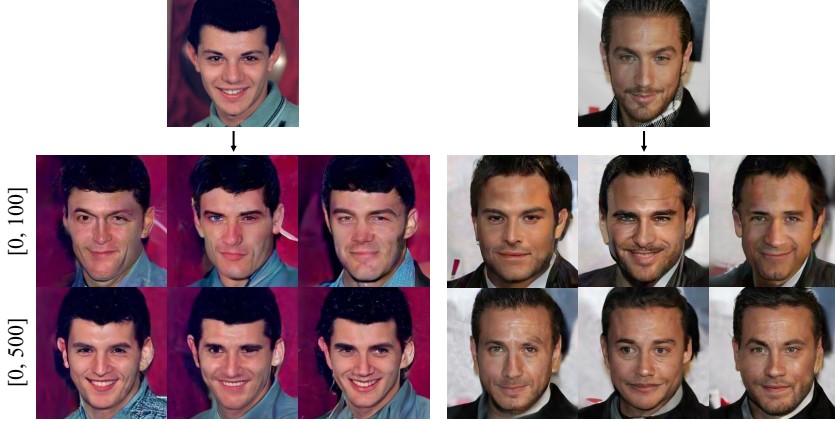

Figure 9: Image variation results. The numbers in parentheses denote the indices of the latent elements fixed, i.e., shared within the generated images at the same rows.

image by solving the forward ODE, edit one element of the feature, and reconstruct the image by solving the reverse ODE.

**Image variation** Moreover, thanks to a hierarchical structure of latent variables, we can generate diverse variations of an input image, as shown in Fig. 9. The hierarchical representation of GDM allows us to control the degree of variation by selecting which latent groups are fixed. When variables within the range of $[0, 100]$ are fixed, there are considerable differences in the overall appearances. However, fixing variables within the range of $[0, 500]$ results in only slight variations in the finer details.

## 5   LIMITATIONS AND CONCLUSION

Our proposed diffusion model with group-wise noising/denoising scheme connects diffusion with autoregressive and cascaded models, revealing new design choices of the models such as grouping strategy and order of generation. GDM can be extended to the frequency domain, leading to applications in hierarchical representation learning, identifying independent factors of variation, and image editing.

One limitation of GDM is that as the number of groups $k$ increases, sampling efficiency declines. In an extreme case where $k$ equals data dimension $d$, it requires at least $d$ sampling steps as in autoregressive models. This shows the effectiveness of diffusion models over autoregressive models on data types like images, where the information is highly redundant, thus some elements can be easily generated in parallel.

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

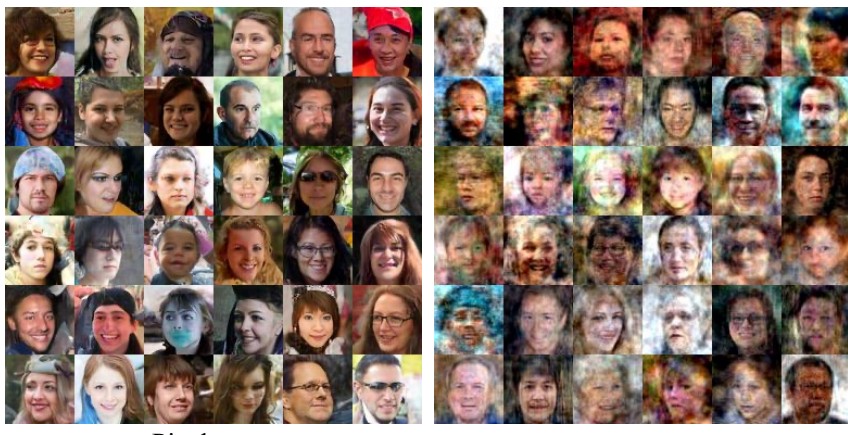

Pixel space                        Frequency space

Figure 10: Synthesis results of GDM-F when training and sampling are done in pixel space (left) and frequency domain (right).

## A APPENDIX

### A.1 TRAINING AND SAMPLING OF GDM-F

The training objective of GDM-F is the same as GDM but defined in the frequency domain:

$$\min_{\boldsymbol{\theta}} \mathbb{E}[||(\bar{\boldsymbol{x}} - \bar{\boldsymbol{z}}) - \bar{\mathbf{u}}_{\boldsymbol{\theta}}(\boldsymbol{x}_t(\bar{\boldsymbol{x}}, \bar{\boldsymbol{z}}), \mathbf{A}(t))||_2^2] \tag{12}$$

Similarly, the generative ODE is

$$d\bar{\boldsymbol{z}}_t = \mathbf{A}'(t)\bar{\mathbf{u}}_{\boldsymbol{\theta}}(\bar{\boldsymbol{z}}_t, t)dt. \tag{13}$$

The final samples are obtained by $\boldsymbol{z}_0 := \mathbf{U}\bar{\boldsymbol{z}}_0$. However, commonly used neural network architectures like convolutional networks are not designed to handle the frequency domain inputs and outputs. Therefore, we perform training and sampling in the pixel space for better sample quality. With $\mathbf{u}_{\boldsymbol{\theta}}(\boldsymbol{x}_t(\boldsymbol{x}, \boldsymbol{z}) = \mathbf{U}\bar{\mathbf{u}}_{\boldsymbol{\theta}}(\boldsymbol{x}_t(\bar{\boldsymbol{x}}, \bar{\boldsymbol{z}}), \mathbf{A}(t))$, we have

$$\min_{\boldsymbol{\theta}} \mathbb{E}[||(\bar{\boldsymbol{x}} - \bar{\boldsymbol{z}}) - \bar{\mathbf{u}}_{\boldsymbol{\theta}}(\boldsymbol{x}_t(\bar{\boldsymbol{x}}, \bar{\boldsymbol{z}}), \mathbf{A}(t))||_2^2] \tag{14}$$

$$= \mathbb{E}[||(\mathbf{U}^T(\boldsymbol{x} - \boldsymbol{z} - \mathbf{u}_{\boldsymbol{\theta}}(\boldsymbol{x}_t(\boldsymbol{x}, \boldsymbol{z}), \mathbf{A}(t))))||_2^2] \tag{15}$$

$$= \mathbb{E}[||(\boldsymbol{x} - \boldsymbol{z}) - \mathbf{u}_{\boldsymbol{\theta}}(\boldsymbol{x}_t(\boldsymbol{x}, \boldsymbol{z}), \mathbf{A}(t))||_2^2]. \tag{16}$$

We integrate the following generative ODE backward:

$$d\boldsymbol{z}_t = \mathbf{U}\mathbf{A}'(t)\mathbf{U}^T\mathbf{u}_{\boldsymbol{\theta}}(\boldsymbol{z}_t, t)dt \tag{17}$$

As shown in Fig. 10, better sample quality is indeed achieved when training and sampling are done in pixel space rather than in the frequency domain.

### A.2 THE NUMBER OF GROUPS

Fig. 11 shows the FID curves with respect to the number of score function evaluations. We use the right $\rightarrow$ left noise schedule, varying the number of groups $k$ from 1 to 29. We conclude that as the number of groups grows, we can obtain more controllability as shown in Sec. 4.3, but at the cost of the sampling efficiency.

### A.3 OPTIMAL TIME INTERVAL FOR SYNTHESIS QUALITY

For latent groups $S_1 = \{0, ..., 29\}, S_2 = \{30, ..., 99\}, S_3 = \{100, ..., 499\}$, and $S_4 = \{500, ..., r^2 - 1\}$, we have to assign time intervals to each group for training and sampling, respectively. As shown

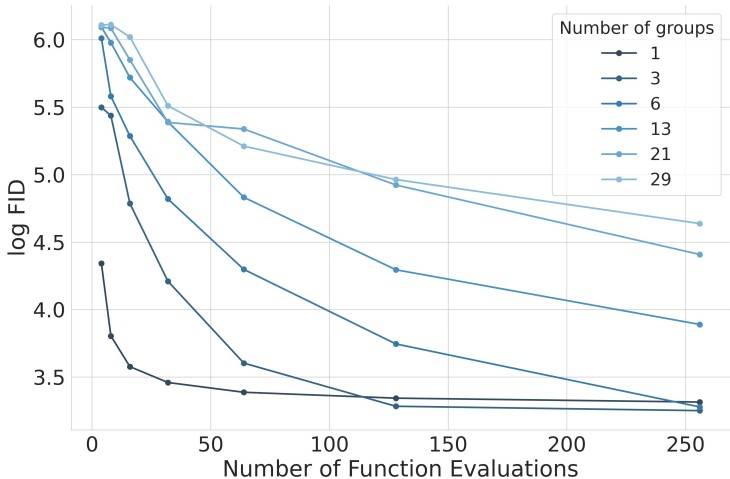

Figure 11: Effect of the number of groups $k$ in FID score. FID is assessed using 5000 samples on the cifar10 dataset.

Table 2: FID10K results for each time interval on FFHQ $64 \times 64$ dataset. Note that the time intervals need to be chosen separately for training and generation time. For instance, the top left cells indicate the FID result when $t_{\text{end}_4}, t_{\text{end}_3}$, and $t_{\text{end}_2}$ are $0.1, 0.2, 0.5$ during training and $0.1, 0.8, 0.9$ during sampling, respectively.

| Training / Generation | 0.1, 0.8, 0.9 | 0.3, 0.8, 0.9 | 0.7, 0.8, 0.9 | 0.1, 0.2, 0.9 | 0.1, 0.5, 0.9 | **0.1, 0.2, 0.3** | 0.1, 0.2, 0.5 |
|---|---|---|---|---|---|---|---|
| 0.1, 0.2, 0.5 | 25.39 | 29.47 | 34.77 | 25.58 | 24.97 | 25.98 | 25.74 |
| 0.3, 0.4, 0.7 | 21.38 | 29.39 | 34.54 | 21.16 | 20.63 | 21.42 | 21.23 |
| 0.5, 0.6, 0.9 | 21.48 | 31.68 | 36.16 | 20.74 | 20.40 | 20.66 | 20.62 |
| **0.6, 0.8, 0.9** | 19.69 | 27.75 | 32.83 | 17.60 | 17.76 | **17.41** | 17.55 |
| 0.7, 0.8, 0.9 | 22.59 | 34.21 | 39.61 | 20.72 | 20.83 | 20.31 | 20.42 |

in Tab. 2, the best result is obtained when the high-frequencies are emphasized in training time and the low frequencies are emphasized in generation time. This is an interesting observation that contrasts with previous work, where the best results are obtained when coarse aspects are given more weight during training (Ho et al., 2020; Choi et al., 2022), and fine details are emphasized in sampling time (Dhariwal & Nichol, 2021).

## A.4 FREQUENCY BASES

So far, we have not specified the choice of the orthogonal matrix $\mathbf{U}$ for GDM-F. Fig. 12 shows examples of Gaussian blur basis (Lee et al., 2022) used in our experiments. Specifically, Gaussian blur basis $\tilde{\mathbf{U}}$ satisfies $\mathbf{W} = \tilde{\mathbf{U}} \mathbf{D} \tilde{\mathbf{U}}^T$ where $\mathbf{W}$ is a Gaussian blurring matrix. Note that other frequency bases like discrete cosine basis used in Hoogeboom & Salimans (2022); Rissanen et al. (2022) are equally applicable.

## B IMPLEMENTATION DETAILS

Throughout our experiments, we use DDPM++ architecture (Song et al., 2020) from the code of Karras et al. (2022)[1]. We use their script for computing FID, and most of the training and model configurations are also adapted from Karras et al. (2022). The random seed is fixed to $0$ in all experiments except those that require multiple runs. We linearly anneal the learning rate as in previous work (Karras et al., 2022; Song et al., 2020).

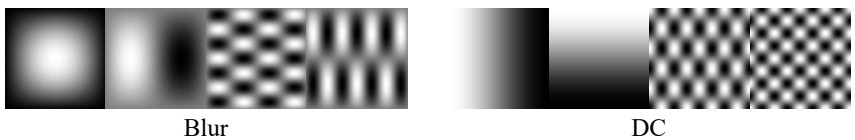

Figure 12: Examples of blur and discrete cosine (DC) bases.

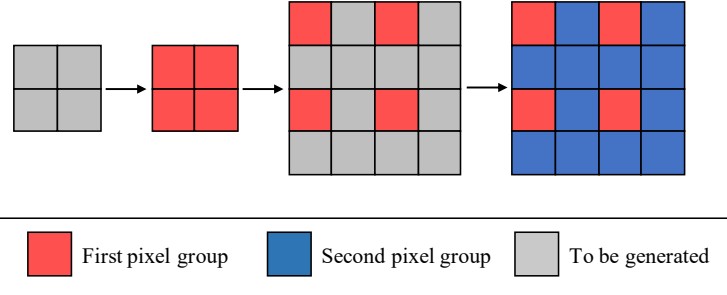

Figure 13: Generation process of the cascaded diffusion models with $2\times$ scale factor.

## C  RELATIONSHIP WITH OTHER MODELS

### C.1  GDM GENERALIZES AUTOREGRESSIVE MODEL

For $\boldsymbol{x} \in \mathbb{R}^d$, AR models define a generative model $p_{\boldsymbol{\theta}}(\boldsymbol{x}) = \prod_{i=1}^{d} p_{\boldsymbol{\theta}}(\boldsymbol{x}_i | \boldsymbol{x}_{<i})$. While a mixture of logistic distributions is often used to model $p_{\boldsymbol{\theta}}(x_i | x_{<i})$, we define it as Gaussian distribution $p_{\boldsymbol{\theta}}(\boldsymbol{x}_i | \boldsymbol{x}_{<i}) = \mathcal{N}(f_{\boldsymbol{\theta}}(x_{<i}), \sigma^2 I)$, where $f_{\boldsymbol{\theta}}(\cdot)$ is a neural network, and $\sigma$ is a constant. AR models are trained by maximizing

$$\max_{\boldsymbol{\theta}} \mathbb{E}_{\boldsymbol{x}} \mathbb{E}_{i \sim \mathcal{U}\{1,d\}} [-\log p_{\boldsymbol{\theta}}(\boldsymbol{x}_i | \boldsymbol{x}_{<i})] \tag{18}$$

or equivalently, minimizing

$$\min_{\boldsymbol{\theta}} \mathbb{E}_{\boldsymbol{x}} \mathbb{E}_{i \sim \mathcal{U}\{1,d\}} [\frac{1}{\sigma^2} ||\boldsymbol{x}_i - f_{\boldsymbol{\theta}}(\boldsymbol{x}_{<i})||^2]. \tag{19}$$

With a slight abuse of notation, for $i \in \{1, ..., d\}$, we denote $i$-th element of $\boldsymbol{x}$ as $\boldsymbol{x}_i$. We will show that we can construct an instance of GDM such that the training objective and sampling procedure are equivalent to the AR model defined above.

Consider GDM where the number of latent groups $k$ is equal to $d$, and the number of steps $N_j$ for each group $j$ is equal to 1. Since $N_j = 1$, we do not need to sample $t$ from the continuous time interval during training. Instead, we uniformly sample an integer $l$ from $\{1, ..., d\}$ and set $t = \frac{l}{d}$. Let us assign the time intervals uniformly to each group, i.e., $t_{\text{start}_j} = (j-1)/d$, and $t_{\text{end}_j} = j/d$. Using the definition of $\mathbf{A}(t)$ in Eq.(10), we have

$$\mathbf{A}(\frac{l}{d})_{jj} = \begin{cases} 0, & (j \leq l) \\ 1, & (j > l) \end{cases} \tag{20}$$

$$\mathbf{A}'(\frac{l}{d})_{jj} = \begin{cases} -d, & (j = l) \\ 0, & (j \neq l) \end{cases} \tag{21}$$

. Since $(\frac{\partial \boldsymbol{x}_t}{\partial t})_j = 0$ for $j \neq l$, Eq. (7) becomes

$$\min_{\boldsymbol{\theta}} \mathbb{E}_{\boldsymbol{x},\boldsymbol{z}} \mathbb{E}_{l \sim \mathcal{U}\{1,d\}} [||(\boldsymbol{z}_l - \boldsymbol{x}_l)d - g_{\boldsymbol{\theta}}(\boldsymbol{x}_t, l)||_2^2], \tag{22}$$

---

[1]https://github.com/NVlabs/edm

with a scalar-valued function $g_{\boldsymbol{\theta}}(\boldsymbol{x}_t, l) \triangleq \boldsymbol{v}_{\theta}(\boldsymbol{x}_t, \mathbf{A}(t))_l$.

As $\boldsymbol{x}_t = \mathbf{A}(t)\boldsymbol{x} + (1 - \mathbf{A}(t))\boldsymbol{z}$, $(\boldsymbol{x}_t)_{\leq l} = \boldsymbol{z}_{\leq l}$, and $(\boldsymbol{x}_t)_{>l} = \boldsymbol{x}_{>l}$. Define $x_{\theta}(\boldsymbol{x}_t, l)$ such that $g_{\boldsymbol{\theta}}(\boldsymbol{x}_t, l) = ((\boldsymbol{x}_t)_l - x_{\boldsymbol{\theta}}(\boldsymbol{x}_t, l))d$. Then we have

$$\min_{\boldsymbol{\theta}} \mathbb{E}_{\boldsymbol{x},\boldsymbol{z}} \mathbb{E}_{l\sim\mathcal{U}\{1,d\}} [||(\boldsymbol{z}_l - \boldsymbol{x}_l)d - ((\boldsymbol{x}_t)_l - x_{\boldsymbol{\theta}}(\boldsymbol{x}_t, l))d||_2^2] \tag{23}$$

$$= \mathbb{E}_{\boldsymbol{x},\boldsymbol{z}} \mathbb{E}_{l\sim\mathcal{U}\{1,d\}} [||(\boldsymbol{z}_l - \boldsymbol{x}_l)d - (\boldsymbol{z}_l - x_{\boldsymbol{\theta}}(\boldsymbol{x}_t, l))d||_2^2] \tag{24}$$

$$= \mathbb{E}_{\boldsymbol{x},\boldsymbol{z}} \mathbb{E}_{l\sim\mathcal{U}\{1,d\}} [d^2||\boldsymbol{x}_l - x_{\boldsymbol{\theta}}(\boldsymbol{x}_t, l)||_2^2] \tag{25}$$

$$= \mathbb{E}_{\boldsymbol{x}} \mathbb{E}_{l\sim\mathcal{U}\{1,d\}} [d^2||\boldsymbol{x}_l - x_{\boldsymbol{\theta}}(\boldsymbol{x}_{>l}, l)||_2^2] \tag{26}$$

The last equality holds because $(\boldsymbol{x}_t)_{\leq l} = \boldsymbol{z}_{\leq l}$ is an independent noise and therefore not needed to predict $\boldsymbol{x}_l$, and $(\boldsymbol{x}_t)_{>l} = \boldsymbol{x}_{>l}$. We can see that Eq. (26) is equivalent to Eq. (19).

Since the $l$-th element of $\boldsymbol{v}_{\boldsymbol{\theta}}(\boldsymbol{x}_t, \mathbf{A}(t))$ is $g_{\boldsymbol{\theta}}(\boldsymbol{x}_t, l) = ((\boldsymbol{x}_t)_l - x_{\boldsymbol{\theta}}(\boldsymbol{x}_{>l}, l))d$ and 0 otherwise and the step size is $1/d$, Eq. (9) leads to the following sampling procedure:

- Initialize $\boldsymbol{x} \in \mathbb{R}^d$ with noise from $\mathcal{N}(\mathbf{0}, \mathbf{I})$.
- For $l$ in $d, ..., 1$,
    - Update $\boldsymbol{x}_l \leftarrow x_{\boldsymbol{\theta}}(\boldsymbol{x}_{>l}, l)$

Note that the initialization does not affect the final samples as $x_{\boldsymbol{\theta}}(\boldsymbol{x}_{>l}, l)$ is a function of previously generated elements $\boldsymbol{x}_{>l}$. At a high level, this instantiation of GDM is trained to predict one element of data from given elements at each step. In sampling time, it generates data from one element at each time using one sampling step, as in AR models. So far, we have not specified how each $i$ is assigned to each $j$, which determines the order of generation. This formulation encompasses all possible orders that need to be specified by practitioners as in AR models.

## C.2 GDM GENERALIZES CASCADED DIFFUSION MODELS

Cascaded diffusion models generate high-dimensional data more effectively by splitting them into multiple groups and synthesizing them in a step-by-step manner. Here, we only consider CDMs with $2\times$ scale factor and two training stages for simplicity. We also assume that no low-pass filtering is used before subsampling in the first training stage of CDMs.

Fig. 13 depicts the generation process of CDMs on image data. In the first stage, CDMs generate the first pixel group indicated by red color. In the second stage, the generated images are upsampled, and the second group of pixels is generated without changing the value of the first pixel group. This is a consequence of not using low-pass filtering during the first training stage. Since two groups of pixels are generated sequentially, this exact behavior can be also modeled by our method (in this case, $k = 2$).

It is noteworthy that the results of GDM with CDM grouping strategy cannot be directly compared to standard CDMs in an apple-to-apple manner because of several differences in implementation. First, low-pass filtering is in fact applied before downsampling images in many cases.[2] Moreover, ad-hoc data augmentation techniques are often used in CDMs to reduce the train/test mismatch. Finally, CDMs have separate models for each stage while we only use a single model.

## D RELATED WORK

**Order-agnostic autoregressive models** Once GDM is trained on multiple orderings and grouping strategies, it can generate data in any given ordering and grouping strategy it is trained on. Since GDM is a generalization of AR models, it has a connection between order-agnostic autoregressive models (Germain et al., 2015; Uria et al., 2014). GDM also shares a similarity with Autoregressive Diffusion Models (ARDM) Hoogeboom et al. (2021), especially when $k = d$ and $N_j > 1$. The difference is that 1) GDM operates on continuous data while ARDM handles discrete data using discrete diffusion (Austin et al., 2021), and 2) GDM is not necessarily restricted to generating one element at once in contrast to ARDM.

---

[2]This is the default behavior of many libraries including `torchvision`.

**Image editing** There are several studies that utilize diffusion models for image editing. Kwon et al. (2022) and Tumanyan et al. (2023) manipulate the intermediate feature maps of the U-Net for image editing. While they try to discover the semantically meaningful directions in the feature space of the pre-trained diffusion models, GDM allows us to assign an interpretable meaning to each group of the initial noise in advance by specifying the group-wise noise schedule.

