# OpenReview forum: "Sequential Data Generation with Groupwise Diffusion Process"
_ICLR.cc/2024/Conference — Submitted to ICLR 2024_

### Official Review · Reviewer_efZH · 2023-10-31

**Soundness:** 2 fair
**Presentation:** 1 poor
**Contribution:** 3 good
**Rating:** 5
**Confidence:** 4

**Summary:**

This work proposes a Groupwise Diffusion Model (GDM), which uses a new type of forward process, for the interpolation between data $\boldsymbol{x}$ and noise $\boldsymbol{z}$ by dividing data into multiple groups and diffusing one group at once. The authors provide an extension to the frequency domain termed as GDM-F. The work claims that a notable characteristic of GDM is that the latent space now possesses group-wise interpretable meaning, for instance to attribute changes in features of an image.

**Strengths:**

Originality: To the best of my knowledge the work seems original, based on the groupwise strategy as an alternative forward process in diffusion models.

Quality: The experiments presented in the work look convincing.

Clarity: The work does not properly explain the methodology of the groupwise strategy which it is the key point in the work. There are some problems in the mathematical notations and inconsistencies.

Significance (importance): The work presents interesting experiments that might be quite useful for the practitioners in the field. There is an interesting contribution regarding a more controlled interpretability when using diffusion models.

**Weaknesses:**

-The the work lacks of an appropriate connection between methodology and experiments that lead the reader to a better understanding of the work.

-The work does not properly explain the methodology of the groupwise strategy which it is the key point in the work. There are some problems in the mathematical notations and inconsistencies.

-There is a problem in the way the results are presented, the figures are cited and appear inappropriately along the document.

**Questions:**

---Specific comments---

-Shouldn't the title be "Sequential Data Generation with Group-wise Diffusion Processes" or "Sequential Data Generation with a Group-wise Diffusion Process"?

-After Eq. (1): It appears a variable "$a$" instead of "$\alpha$", $a(0)=1$ and $a(1)\approx 0	$, is it a typo? Or what is $a$?

-Before Eq. (2): correct singular or plural in "some recent work instead use the linear interpolation", "...recent work uses the linear" or "...recent works use"?

-In Eq. (3): the variable $\boldsymbol{x_{\theta}}$ was not introduced. Also, what is $\boldsymbol{\theta}$ in the minimization argument? It was not introduced.

-Before Eq. (4): There is a double notation for the variable $\boldsymbol{v_{\theta}}$ or $\mathbf{v_{\theta}}$, or are they different?

-Include a comma "," after Eq. (2) and Eq. (3).

-Section 3.1: I believe the first paragraph should be rewritten in a more detailed way. This paragraph is key to lead the reader to understand the way data division in performed. For instance, it is not clear "we divide
data into $1 \leq k \leq d$ groups", does this mean we divide the data into any possible integer in the interval $1 \leq k \leq d$, then that integer value represents the number of groups? Or is it that you want to divide data into a number of $d$ groups, where each group has a length of $k$? Shouldn't we refer to the total number of data and then refer to the splits of that number as per your method proposes?

-Also Section 3.1: It suddenly appears "we divide the indices $\{1, ..., d\}$ into a partition $\{S j \}^k_{j=1}$". Do you divide the set $\{1, ..., d\}$ into another set $\{S_1,S_2...,S_k\}$? So, what is the meaning or definition of each $S_i$?
That is not clear to me.

-At the end of Section 3.1: It reads "the elements of j-th latent group are diffused into noise", what is a latent group? When was the name latent group introduced? Or what variable corresponds to such a latent group?

-Before Eq. (8): It reads "Instead, we find that it is beneficial to define $\mathbf{u_{\theta}}$", why is it beneficial? It is important to include the explanation in the phrase to guide the reader.

-There is no consistence with the figures when they are cited in the text and when they appear. See for instance Fig. 5 and Fig. 4.

-In the experiments section: It might be quite important to be able to interconnect what it was derived in the methodology with the experiments. For instance where it reads "Fig. 6 shows that manipulating a single element in the latent vector of the lowest
frequency band results in a change in a single high-level attribute of images.", where is that latent vector appearing in the methodology, what Equation? how can the reader understand from the methodology where the variable that seems quite important is appearing. What is the dimensionality of such a variable?

-Introduce the acronym FID!

-Include comma "," after Eq. (2) and (3).

-Period "." after Eq. (8).

---Other Questions---

-In the Fig. 6 where traversing a single latent variable is associated to a particular featured changed in the images, what is the effect of traversing other elements like (2), (3), (10) or (20)?

-In the practice, in which scenarios should we apply GDM or GDM-F?

---

> ### Author Response · Authors · 2023-11-17
>
> We appreciate the reviewer's constructive and detailed comments. Especially, our strengths recognized by the reviewer are:
> - Originality of the group-wise strategy
> - Convincing experiments
> - Controllability
>
> In the following, we answer the specific questions raised by the reviewer.
>
> ### Presentation-level comments
> Thank you for pointing out the typos, grammatical errors, and other presentation-level issues. They were very helpful. We reflected them in our revised version, so please refer to it.
>
> ### Questions about grouping (Sec. 3.1.)
>
> > ...does this mean we divide the data into any possible integer in the interval $1 \leq k \leq d$, then that integer value represents the number of groups?
>
> This is right. We revised the paragraph as follows:
> > ...we divide data $\boldsymbol x \in \mathbb R^d$ into $k\in \{1,...,d\}$ disjoint groups and assign different noise schedules for each group.
> Specifically, we divide the indices $\{1,...,d\}$ into a partition $\{S_j\}_{j=1}^k$ where $\sum_j |S_j|=d$. $S_j$ is used for determining membership of $\boldsymbol x_i$ to $j$-th group by testing $i \in S_j$.
>
> For example, in the block-wise grouping, $j$-th group corresponds to the $j$-th block of an image (there are $k$ blocks in total). $S_j$ is the set of the indices of each block.
>
> ### The latent group
>
> > What is a latent group? When was the name latent group introduced? Or what variable corresponds to such a latent group?
>
> Throughout the paper, "group of latent variables" and "latent group" were used interchangeably. In the revised version, we clarified this in the last paragraph of the introduction. Also, there was a typo in Sec. 3.1 which we replaced "latent group" with "group" during the revision. Thank you for pointing out.
>
>
> ### Why do we use $\boldsymbol u_{\boldsymbol \theta}$ instead of $\boldsymbol v_{\boldsymbol \theta}$?
> The problem of directly predicting $\mathbf A'(t)(\boldsymbol x - \boldsymbol z)$ is that a neural network has to infer the derivative of $\mathbf A(t)$ from $\mathbf A(t)$. On the other hand, $\boldsymbol u_{\boldsymbol \theta}$ parameterization does not face such challenges since the derivative of $\mathbf A(t)$ is already given as a part of it. In practice, we tried both and found that $\boldsymbol u_{\boldsymbol \theta}$ indeed outperforms $\boldsymbol v_{\boldsymbol \theta}$.
>
> ### Effect of traversing other elements
> > In the Fig. 6 where traversing a single latent variable is associated to a particular featured changed in the images, what is the effect of traversing other elements like (2), (3), (10) or (20)?
>
> We have tried traversing other elements, but in some cases the factors captured are not interpretable by humans. We argue that this is not a limitation of our work in particular but due to the ill-posedness of the problem itself. For example, in the first column of Fig. 1 in $\beta$-VAE paper [1], one can see that changing the azimuth also affects e.g. hair color. Without additional assumptions or inductive biases, it is too ambitious to hope that a factor captured by neural networks is always interpretable by humans because the generative model that transports $p(\boldsymbol z)$ to $p(\boldsymbol x)$ is not unique [2].
>
> ### In which scenarios should we apply GDM or GDM-F?
> Compared to standard diffusion models, GDM can be particularly useful when we have some apriori knowledge of the underlying data-generating process. For instance, in video data, temporal ordering can be a natural design choice, so sequentially generating each frame could be more effective. In this case, the pixels in each frame consist of a group, and $k$ is equal to the number of frames.
>
> One could further consider generating some frames in parallel rather than generating each frame sequentially because the information in neighboring frames would be highly redundant. One could even generate some key-frames first and then interpolate between them, or generate key pixels/frames first and then fill in the rest across the spatial/temporal axis, which has shown to be effective in practice [3]. GDM unifies all these grouping and ordering strategies into a single unified framework.
>
> One might consider applying GDM-F if the most salient features of the data are captured by low-frequency components as in images. If this is the case, GDM-F opens up many interesting applications such as hierarchical latent space, extracting independent generative factors that are semantically meaningful, editing, etc., as we have shown in the paper.
>
>
> [1] β-VAE: LEARNING BASIC VISUAL CONCEPTS WITH A CONSTRAINED VARIATIONAL FRAMEWORK
>
> [2] Challenging Common Assumptions in the Unsupervised Learning of Disentangled Representations
>
> [3] Imagen Video: High Definition Video Generation with Diffusion Models

---

> > ### Comment · Reviewer_efZH · 2023-11-22
> >
> > Thanks to the authors for the different discussion regarding the comments, suggestions and questions.

---

> ### Author Response · Authors · 2023-11-21
>
> Dear Reviewer efZH,
>
> We thank the Reviewer for the constructive comments. As the end of the discussion period is approaching, we would like to ask whether our responses have addressed your concerns and questions adequately. If not, we would be happy to discuss further.
>
> Regards,
>
> the authors.

---

### Official Review · Reviewer_5sF6 · 2023-11-04

**Soundness:** 2 fair
**Presentation:** 2 fair
**Contribution:** 2 fair
**Rating:** 3
**Confidence:** 4

**Summary:**

This paper propose GDM that generalizes the diffusion process by grouping the data into different blocks and applying different schedule to each block. Specifically, it shows that an Bottom→Top scheme to partite the data is able to improve the general generating performance.

**Strengths:**

The groupwise diffusion model is a unified model of some existing works. And the authors take effort to combine this model with other approaches such as frequency domain as well as conducting a lot of experiment to demonstrate the approach.

**Weaknesses:**

1. Overall, I find this method has limited novelty. It is a simple extension of the model such as rectified flow by using a matrix interpolation function.

2. When applied to the real problem, this approach seems adhoc. It is not explained and supported that why we should partite the data by a certain order and why such partition gives better generation quality.

**Questions:**

Can you explain why partite the data by a bottom->top approach is better than others?

---

> ### Author Response · Authors · 2023-11-17
>
> We appreciate the reviewer's comments. The review contains some significant misconceptions, which we will address below.
>
>
> ### Limited novelty
> > Overall, I find this method has limited novelty. It is a simple extension of the model such as rectified flow by using a matrix interpolation function.
>
> Our method uses a very specific interpolation function in Eq. 10, not just any matrix-valued function. The connection between autoregressive models and cascaded diffusion models is only valid because of this specific choice of the interpolation function. To our knowledge, extending diffusion models to support the sequential generation of each group has not been done previously. The other two reviewers also agreed that the connection with autoregressive and cascaded diffusion models is indeed novel.
>
> #### Other contributions
> Our unified model also accompanies useful properties including hierarchical latent space, finding independent generative factors, etc., which have not been explored in previous work.
>
> ### Why do certain grouping and ordering work better than others?
>
> > It is not explained and supported that why we should partite the data by a certain order and why such partition gives better generation quality.
>
>
> The aim of this paper is not to propose the best grouping and ordering strategy but to show that grouping and ordering is important design choices in sequential generative models, and we are already very explicit about this in the paper. The reviewer is correct that the reason why certain grouping and ordering work better than others is often not human-interpretable. Indeed, the progressive growing strategy of CDM is outperformed by bottom->top ordering or even CDM-inverse, and it is hard to interpret the meaningful distinction between the best and worst performing orderings of the block-wise grouping in Fig. 5. This is **not a limitation of our work but is in fact a significant observation** because it implies that the commonly used groupings and orderings, even if they appear to be a good design choice from a perspective of a human, could be not the best choice.

---

> ### Author Response · Authors · 2023-11-21
>
> Dear Reviewer 5sF6,
>
> We thank the Reviewer for the constructive comments. As the end of the discussion period is approaching, we would like to ask whether our responses have addressed your concerns and questions adequately. If not, we would be happy to discuss further.
>
> Regards,
>
> the authors.

---

### Official Review · Reviewer_Mi2L · 2023-11-09

**Soundness:** 3 good
**Presentation:** 3 good
**Contribution:** 3 good
**Rating:** 6
**Confidence:** 4

**Summary:**

The paper focuses on groupwise diffusion process, namely GPM. The key is that it generates one sample at a time for each group. This new framework generalizes to the autoregressive model in certain circumstances. Another benefit is that the latent space becomes interpretable because of the group-wise generative process. The GPM further extends to the frequency domain, allowing each group to encode data at different frequency levels. The authors demontrate several applications like image editing, disentangled image semantic attributes, and generating variations.

**Strengths:**

- The GPM proposed in the papers is quite interesting along with its properties. It can be extended to the frequency domain.
- The demonstrated examples are promising.
- It adds new connections between GDM and certain forms of autoregressive models and cascaded diffusion models.

**Weaknesses:**

- It seems that the number of groups, k, is a hyperparameter where not much discussion is given.
- There are no comparisons to other generative models like VAE or GAN or even guided diffusion.

**Questions:**

1. How do you choose the number of groups?
2. If we set the number of groups equal to 1, what should we expect? How does it compare with the generated data from DDPM or DDIM?

---

> ### Author Response · Authors · 2023-11-17
>
> We appreciate the reviewer's insightful comments, highlighting our strengths in
>
> - interesting connection between diffusion and autoregressive/cascaded diffusion models,
> - intriguing properties of GDM supported by promising experiment results
> - extension to the frequency domain
>
> Regarding your specific concerns:
>
> ### The number of groups $k$
> $k$ can be chosen based on the user's demand. For example, if a user wants autoregressive models, $k$ should be set to $d$. If a user wants cascaded diffusion with two stages, $k$ should be 2. What types of sequential generation a user needs depends on the data type and the user's prior knowledge of it.
>
> Also, setting $k$ to be greater than 1 enables various use cases such as obtaining hierarchical representation, extracting independent generative factors, image editing, and so on.
>
>
>
> ### If we set the number of groups equal to 1, how does it compare with DDPM or DDIM?
> GDM is equivalent to previous diffusion models (or rectified flow) when $k=1$. For example, if we use the interpolation coefficients in Eq. 1, GDM reduces to the variance-preserving (VP) diffusion models. We made this even more clear in the last paragraph of Sec. 3.1. DDPM is a discretization of VP diffusion, and DDIM is a discretization of the probability-flow ODE.
>
> ### Comparison with VAEs, GANs, and guided diffusion
> We would like to emphasize that the primary focus of our paper is to propose a unified framework for diffusion models and other sequential generative models and dissect interesting properties arising from it.
>
> Since we do not claim state-of-the-art generation performance, we believe it is not necessary to compare with VAEs and GANs, which are distinct families from diffusion and autoregressive models. Also, guided diffusion [4] is a method that utilizes classifier gradients for making a diversity-fidelity trade-off, which is not directly aligned with the scope of our work.
>
> When $k=1$, GDM reduces to standard diffusion models, so the sample quality is on par with other diffusion models, given that the same architecture and training configuration are used. For $k>1$ case, please refer to Appendix A.2.
>
> [1] Cascaded Diffusion Models for High Fidelity Image Generation
>
> [2] PixelCNN++: Improving the PixelCNN with Discretized Logistic Mixture Likelihood and Other Modifications
>
> [3] simple diffusion: End-to-end diffusion for high resolution images
>
> [4] Diffusion Models Beat GANs on Image Synthesis

---

> ### Author Response · Authors · 2023-11-21
>
> Dear Reviewer Mi2L,
>
> We thank the Reviewer for the constructive comments. As the end of the discussion period is approaching, we would like to ask whether our responses have addressed your concerns and questions adequately. If not, we would be happy to discuss further.
>
> Regards,
>
> the authors.

---

### Meta-Review · Area_Chair_W47e · 2023-12-09

**Metareview:**

The paper presents a groupwise diffusion process. Two reviewers argued for rejection, while one believed it could be accepted. The idea was thought to be potentially interesting, but concerns were raised about the motivation for the methodology and lack of comparison with other generative models.

**Justification For Why Not Higher Score:**

The reviewers' collectively believe the paper should be rejected.

**Justification For Why Not Lower Score:**

N/A

---

### Decision · Program_Chairs · 2024-01-16

Reject